# Frequency and Prognostic Significance of Intertumoural Heterogeneity in Multifocal Jejunoileal Neuroendocrine Tumours

**DOI:** 10.3390/cancers14163963

**Published:** 2022-08-17

**Authors:** Moritz Jesinghaus, Jelte Poppinga, Bettina Lehman, Elisabeth Maurer, Annette Ramaswamy, Albert Grass, Pietro Di Fazio, Anja Rinke, Carsten Denkert, Detlef K. Bartsch

**Affiliations:** 1Institute of Pathology, Phillips University Marburg and University Hospital Marburg, 35043 Marburg, Germany; 2Department of Visceral, Thoracic and Vascular Surgery, Phillips University Marburg and University Hospital Marburg, 35043 Marburg, Germany; 3Department of Gastroenterology, Phillips University Marburg and University Hospital Marburg, 35043 Marburg, Germany

**Keywords:** multifocal neuroendocrine tumors, SI-NETs, intertumoral heterogeneity

## Abstract

**Simple Summary:**

Well differentiated jejunoileal neuroendocrine tumors (SI-NETs) commonly present as multiple primaries, which are believed to be clonally unrelated. Our study aimed to explore whether these multifocal lesions show intertumoral differences between the most common histomorphological parameters and if so, whether these differences can be associated with patient prognosis. While WHO grade and standard neuroendocrine markers were found to be mostly stable, we observed intertumoral heterogeneity in the expression of SSTR2, CDX2, and serotonin, but observed no survival differences between the expression groups or in comparison to unifocal NETs. Although multifocal SI-NETs showed some degree of heterogeneity in their central morphological parameters, these findings do not appear to be of major clinical significance, rendering an extensive testing of all multifocal lesions as not necessarily required.

**Abstract:**

Background: A recent study found that multifocal jejunoileal neuroendocrine tumors (SI-NETs) are genetically unrelated synchronous neoplasms. So far, it is unclear if this finding of synchronous independent neoplasms is mirrored by heterogeneity of key morphological parameters of SI-NETs and how it affects patient survival. Methods: We separately assessed WHO grade (based on the Ki-67 index), expression of basal diagnostic markers (synaptophysin/chromogranin A/CDX2/serotonin), SSTR2a, and the contexture of the immunogenic microenvironment in 146 separate tumors from 28 patients with multifocal SI-NETs and correlated the results with clinicopathological factors and survival. Results: Synaptophysin and chromogranin A were strongly expressed in all tumors. WHO grade was concordant within all multifocal lesions in more than 80% of cases and the highest grade was usually found in the most advanced primary. Intertumoral expression of serotonin, SSTR2, and CDX2 was discrepant in 32%, 43%, and 50% of all patients, respectively. Neither heterogeneity of any of the aforementioned markers nor multifocality itself had any impact on patient survival (*p* = n.s.). Discussion: Multifocal SI-NET show considerable variability in some of the central diagnostic parameters. However, neither intertumoral heterogeneity of those parameters nor multifocality itself had any impact on patient survival, showing that extensive testing of all multifocal lesions is not necessarily required.

## 1. Introduction

Well-differentiated neuroendocrine tumors of the aboral small intestine (SI-NETs) most commonly arise in the distal jejunum and the ileum and originate from serotonin-producing enteroendocrine cells of the small intestinal epithelium [1,2,3,4]. Despite being mostly diagnosed in advanced local and often metastasized stages, SI-NETs usually show a prolonged disease course with high disease-specific survival rates [5]. According to the WHO, the prognosis of SI-NETs generally depends on UICC stage and tumor grade, which divides NETs into three grades based on their proliferative activity [6,7]. Somatostatin-receptor imaging (SRI), especially of subtype 2a (SSTR2a), is an important cornerstone for clinical staging and follow up of SI-NETs and correlates with SSTR2a expression on immunohistochemistry [8,9,10,11,12].

Although SI-NETs are classically sporadic neoplasms that arise outside of defined hereditary syndromes, approximately one third of SI-NETs are multifocal, presenting with multiple separate primary tumors (reported range: 2–100) of variable size and local extension [1,13,14,15,16]. Whole-genome-sequencing data from a recent study by Elias and colleagues on the genetic relationship between multifocal SI-NETs found no intertumoral mutational overlap between the synchronous primaries [17]. These novel findings suggest that multifocal SI-NETs are a polyclonal disease, in which the patients deal with multiple genetically unrelated primary tumors rather than a single monoclonal metastatic neoplasm. In the light of these results, the question arises as to whether this genetic heterogeneity is also mirrored on a morphological level between the multiple primaries, especially in therapy-relevant parameters such as tumor grade or expression of SSTR2a and how these results might influence current diagnostic algorithms of multifocal SI-NETs.

In order to answer this question, we performed an in-depth morphological analysis of intertumoral heterogeneity in a large series of multifocal SI-NETs. Analyzing a total of 146 separate primary tumors from 28 patients, we comparatively assessed proliferative activity/tumor grade (WHO-grade), expression of basal diagnostic markers (synaptophysin, chromogranin A, CDX2, and serotonin), and SSTR2a status as well as the contexture of the immunogenic microenvironment (T-/B-cell density and PD-L1 expression) between the separate tumors. Finally, we analyzed the clinical course of multifocal NETs with and without morphological heterogeneity compared to unifocal jejunoileal neuroendocrine tumors.

## 2. Materials and Methods

### 2.1. Multifocal Ileal Neuroendocrine Tumour Cohort

Twenty-eight patients suffering from multifocal SI-NETs who underwent surgical resection between 2012 and 2021 at the University Hospital Marburg were included in this study. All patients had multifocal SI-NETs, meaning that at the least, two spatially distinct primary tumors were diagnosed in the post-operative pathology report. The number of distinct ileal primaries ranged from 2 to 36 individual tumors per patient (median: 6 primary tumors), resulting in a total of 212 separate NETs that were diagnosed in the initial post-operative diagnostic workup. Formalin-fixed paraffin embedded tissue of 146 of those 212 tumors was available for this study. Twenty-seven patients had tumors located in the ileum, one patient had jejunal primaries.

During the routine clinicopathological workup after the resection, the locally most advanced tumor was determined as the “leading primary NET” and was evaluated regarding it’s Ki-67 index in order to determine the tumor grade according to WHO criteria (G1: <3%, *n* = 22; G2: 3–20%, *n* = 5; G3: >20%, *n* = 1), expression of SSTR2a, and serotonin. The leading primary NET served as the reference for our comparative analyses of all multifocal tumors.

NETs from other locations or neuroendocrine carcinomas as well as cases with incomplete clinicopathological/survival data, missing patient consent, or insufficient tissue were excluded. Survival data as well as clinicopathological characteristics from all patients were extracted from internal hospital records. Survival data from of some these patients were part of a previously published collective [18].

The local ethics committee of the University Hospital Marburg (reference number: AZ 206/10) approved this study.

### 2.2. Control Cohort of Unifocal Jejunoileal NETs

We compared the survival characteristics of our multifocal NET series with a cohort of 88 advanced unifocal ileojejunal NETs (UICC stage IV), who underwent resection at the University Hospital Marburg during the same time period. Survival data from some of these patients were part of a previously published collective [19].

### 2.3. Tissue-Based Analyses

Full block H&E slides from all primary tumors were rescreened by an experienced GI-pathologist (M.J.) using an Olympus BX46 microscope at the beginning of this study in order to confirm the initial diagnosis. All immunohistochemical analyses were also performed by an experienced GI-pathologist (M.J.) who was blinded to all clinicopathological parameters. For all investigations the most locally advanced NET, which received Ki-67 testing at the time of diagnosis, was taken as the reference for all further analyses (leading primary NET).

#### 2.3.1. Tissue Microarray Construction

Following the initial re-evaluation, representative tumor areas from the tumor center and the invasive front from every suitable primary tumor that harbored a high burden of invasive tumor were marked on HE slides. Next, formalin-fixed paraffin-embedded (FFPE) tumor samples from each separate primary tumor of the multifocal ileum NETs were assembled into a tissue microarray (TMA) using a fully automated Tissue Microarrayer (TMA Grandmaster, Sysmex, Budapest, Hungary) with a core size of 1.5 mm.

#### 2.3.2. Immunohistochemistry

Two µm thick unstained slides from the TMA were stained on a Leica Bond stainer (Leica Biosystems, Wetzlar, Germany) with antibodies against synaptophysin (DAK-SYNAP, Agilent Dako, Santa Clara, CA, USA), chromogranin A (LK2H10, Zytomed, Berlin, Germany), serotonin (5HT-209, Agilent Dako, Santa Clara, CA, USA), PD-L1 (E1L3N, Cell Signaling, Boston, MA, USA), CD3 (F7.2.38, Agilent Dako, Santa Clara, CA, USA), CD20 (L26, Agilent Dako, Santa Clara, CA, USA), and CDX-2 (DAK-CDX2, Agilent Dako, Santa Clara, CA, USA). Specific external control tissues for each antibody were deposited as a staining reference on each slide of the TMA.

Two µm thick unstained full-block slides from FFPE blocks were stained with a Ki-67 antibody (MIB-1, Agilent Dako, Santa Clara, CA, USA) and an SSTR2a antibody (polyclonal, Zytomed, Berlin, Germany). For Ki-67, the proliferation zone of the basal epithelium of the jejunoileal mucosa served as an internal staining control on the respective slide. For SSTR2a normal endocrine pancreatic islets served as staining control.

#### 2.3.3. Comparative Analysis of Standard Diagnostic Markers in Multifocal Ileum NETs

The expression of synaptophysin, chromogranin A, CDX2, and serotonin was evaluated on the TMA for each separate primary tumor. The percentage of positive cells and the extent of the positivity was noted, NETs with a block-like expression in the majority of cells were classified as positive, while NETs with an expression in scattered single cells (<5%) were categorized as focally positive/negative (further referred to as negative).

#### 2.3.4. Comparative Analysis of WHO Grade in Multifocal SI-NETs

The proliferative activity of each lesion was assessed on full block slides and each tumor was graded based on its Ki-67 index according to the WHO criteria (G1: <3%, G2: 3–20%, G3: >20%). All slides were at first assessed in scanning magnification (2×) in order to identify the “hot-spot” with the highest proliferative activity for each lesion. Next, the hot-spot was then investigated using the 20× objective and all tumor cells were counted, followed by an additional count of all cells with a positive Ki-67 labeling. In cases where a borderline proliferation activity between different grades resulted from the initial count under the microscope, a camera-captured image of the respective region was taken, printed, and then manually assessed, as described previously [20].

#### 2.3.5. Comparative Analysis of SSTR2a Expression in Multifocal SI-NETs

Expression of SSTR2a was evaluated on full-block slides and was initially classified into four different categories according to the HER2 scoring scheme as proposed by Kasajima et al. (Score 0: absence or weak incomplete membrane staining in <10% tumor cells, Score 1+: Faint/barely perceptible or weak incomplete membrane staining in ≥10% tumor cells; Score 2+: Weak to moderate complete membranous or cytoplasmic staining in ≥10% of tumor cells; Score 3+: Strong complete membranous or cytoplasmic staining in ≥10% of tumor cells) [9]. For further analyses all tumors with an SSTR2a 2+/3+ score were assigned to the SSTR2a-positive category, while SSTR2a 1+ as well as completely negative NETs were assigned to the SSTR2a-low/negative category (further referred to as negative).

#### 2.3.6. Comparative Analysis of the Tumor Microenvironment in Multifocal SI-NETs

The analysis of the tumor microenvironment was evaluated on the TMA for each separate primary tumor. The tumor area occupied by CD3+ T-lymphocytes and CD20+ B-lymphocytes (stromal and intraepithelial) was assessed for each primary tumor. Expression of PD-L1 was separately assessed for tumor cells and immune cells on the TMA [21].

### 2.4. Statistics

Statistical analyses were performed using SPSS version 28 (SPSS Institute, Chicago, IL, USA) using Χ^2^ test as well as Χ^2^ test for trends and Fisher’s exact test. Where necessary, the Bonferroni method was used to correct for multiple testing [22]. Univariate survival analyses were performed using the Kaplan–Meier method and a log-rank test was used to assess the significance of survival differences. The Cox proportional hazard model was used for multivariate analyses. All statistical tests were performed two-sided, *p*-values ≤ 0.05 were considered significant.

## 3. Results

### 3.1. Cohort Description

The median follow-up time of the 28 patients with multifocal SI-NETs was 38.5 months.

Median patient age at the time of resection was 63 years (range: 27–79 years). Fifteen patients were male (54%), 13 patients were female. Nearly all patients showed advanced post-operative tumor stages (according to the Union for International Cancer Control; UICC) with frequent nodal involvement and distant metastases (UICC stage II: 1/28, 4%; stage III: 4/28, 14%, stage IV: 23/28, 81%). Eight patients developed a carcinoid syndrome. Five of the 28 patients died of their disease, no non-tumor-specific deaths were reported. Progress was noted in ten patients. The detailed clinicopathological characteristics of the multifocal cohort are given in Appendix A.

The control cohort of unifocal SI-NETs comprised 88 advanced jejunoileal NETs (UICC stage IV), who underwent resection at the University Hospital Marburg during the same time period. Median patient age in the unifocal cohort was 60 years (range: 28–92 years). Fifty-two patients were male (59%), 36 patients were female. Forty-six patients died during the follow-up time (deaths of any cause, median follow-up time: 48.5 months). The detailed clinicopathological characteristics of the unifocal control cohort are given in Appendix A.

### 3.2. Expression of Standard Diagnostic Markers in Multifocal SI-NETs

Synaptophysin and chromogranin A were diffusely expressed by all tumors (Appendix A), with no significant intertumoral differences for those basal diagnostic markers of SI-NETs. A divergent expression of serotonin between the leading primary NET and synchronous multifocal tumors was observed in 9/28 patients (32%, range: 1–20 divergent tumors). Four of those nine patients showed a serotonin-negative leading primary NET with smaller serotonin-positive NETs, while five patients had a serotonin-positive leading primary NET accompanied by serotonin-negative smaller NETs (Figure 1 and Appendix A). A divergent expression of CDX2 between the leading primary NET and the multifocal tumors was observed in 14/28 patients (50%, range: 1–7 divergent tumors). Five of those fourteen patients showed a CDX2-negative leading primary NET with smaller CDX2-positive NETs, nine patients had a CDX2 positive primary NET accompanied by CDX2-negative smaller NETs (Figure 1 and Appendix A).

### 3.3. Comparative Analysis of WHO Grade in Multifocal SI-NETs

As depicted in Figure 1 and Figure 2, 22 (78%) of the 28 leading primary NETs were G1 neoplasms, 5 (19%) were G2 tumors, and one leading primary NET was highly proliferative and was therefore diagnosed as G3 (4%). In 82% of all patients (23/28) all synchronous NETs showed the same WHO grade (G1: *n* = 21, G2: *n* = 2). A discrepant WHO grade was observed in five patients (18%, range: 1–24 divergent tumors), mostly meaning that a G2/G3 leading primary NET was accompanied by smaller synchronous G1 NETs. In one patient only, we observed a higher grade in a smaller synchronous NET (G2, proliferation rate: 3.5%) as in the leading primary NET (G1, proliferation rate: <1%).

### 3.4. SSTR2a Expression in Multifocal SI-NETs

As depicted in Figure 1 and Figure 3, 19 (68%) of the 28 leading primary NETs fell into the SSTR2 2+/3+ group and 9 (32%) were SSTR2 0/1+. A discrepant SSTR2 status was observed for 43% of the patients (range: 1–10 divergent tumors), mostly meaning that an SSTR2 2+/3+ leading primary NET was accompanied by smaller synchronous SSTR2 0/1+ NETs.

### 3.5. Contexture of the Immunogenic Microenvironment in Multifocal SI-NETs

As shown in Appendix A, multifocal SI-NETs were pauci-immune neoplasms with only scattered tumor-infiltrating lymphocytes (mean T-cell density: 2% of the tumor area; mean B-cell density: <1% of the tumor area), almost all of the tumors were completely negative for PD-L1 with only one tumor showing microfocal positivity (<1% positive tumor cells). No significant differences regarding T-/B-cell density and PD-L1 expression were observed in any of the patients (*p* = n.s.).

### 3.6. Association of Morphological Heterogeneity with Survival

Within the cohort of multifocal NETs, none of a change of WHO grade, SSTR2 status, CDX2, or serotonin expression showed any association with disease progression or patient survival (*p* = n.s.).

In the next step, we investigated the survival impact of our data regarding multifocal SI-NETs in a pooled analysis of 116 patients (28 multifocal SI-NETs and 88 unifocal SI-NETs). We did not observe any significant differences in overall survival between uni- and multifocal SI-NETs (*p* = 0.14, Figure 4A), with multifocal NETs even trending towards a slightly better general survival. None of heterogeneity of WHO grade (*p* = 0.31, Figure 4B), SSTR2a status (*p* = 0.33, Figure 4C), or serotonin expression (*p* = 0.28, Figure 4D) within multifocal NETs had any impact on patient survival in comparison to multifocal NETs without heterogeneity or unifocal SI-NETs.

## 4. Discussion

A recent comparative whole-exome-sequencing study by Elias et al. [17] has addressed the long-standing question of the intertumoral relationship of multifocal SI-NETs and found no genomic overlap between the synchronous primaries. They concluded that multifocal SI-NETs represent a polyclonal disease of multiple genetically unrelated neoplasms rather than one main primary tumor that has regionally metastasized within the small intestine. These novel findings of intertumoral independence raise the question whether this genomic heterogeneity is mirrored on a morphological level and how it potentially influences clinicopathological diagnostics. There are no established diagnostic guidelines for multifocal SI-NETs that specify how many lesions have to receive additional immunohistochemical analysis, for example for the purpose of grading or the determination of SSTR2 expression.

Our study investigated intertumoral heterogeneity of WHO grade, SSTR2 status, expression of basal diagnostic markers of SI-NET, and the immunogenic microenvironment in a series of multifocal NETs comprising 146 individual neoplasms from 28 patients. We aimed to explore how a possible heterogeneity of those markers might influence diagnostic algorithms and how this impacts patient survival.

Besides pTNM stage, WHO grading based on the Ki-67 index is the clinically most important morphological parameter in SI-NETs [23,24]. We demonstrated this to be concordant in almost 80% of multifocal NET. In almost all of the cases with a discordance between the synchronous primaries, the “leading primary NET” tested during routine pathological workup was a G2/G3 neoplasm accompanied by smaller G1 tumors. An increase of grade in a smaller NET was only observed in one patient with a leading G1 NET and synchronous smaller tumor with a low-proliferative G2 NET (3.5%). Although there was minor variability between the synchronous primaries, assessment of WHO grade on the leading primary appeared to be sufficient to adequately portray the general proliferative activity in multifocal SI-NETs, which is in line with previous observations from Nubere et al. [25].

While the NET-defining, strong, and diffuse expression of the neuroendocrine markers synaptophysin and chromogranin A was observed in all lesions [5], the expression of the intestinal transcription factor CDX2 and the hormone serotonin—both reported to be typically expressed in SI-NET [5,23,26]—showed considerable intertumoral variability. Since the initial diagnosis of SI-NETs is often made on extraintestinal biopsies due to their frequent metastatic spread, this finding has diagnostic implications for the pathologist who has to consider that a metastatic NET—although negative or only very focally positive for CDX2 or serotonin—may still be of jejunoileal origin if a comparable H&E morphology is present.

Although expression of SSTR2 (score 2+/3+) was present in at least one investigated tumor in most patients with multifocal SI-NETs, we observed discordant SSTR2 expression in 43% of patients and two individuals in whom all investigated NETs were SSTR2-negative or only very weakly positive. Thus, the fact that especially in small tumors the radionuclide uptake may be low in the first place as well as the quite common combination of immunohistochemically SSTR2-positive and -negative tumors within the same patients, might yield an explanation for the phenomenon that multifocal SI-NETs are often hard to completely assess by functional somatostatin-receptor imaging [27,28].

In the last part of our analyses, we aimed to explore if there are any prognostic differences between multifocal SI-NETs with or without morphological heterogeneity compared to one another and to unifocal SI-NETs. Interestingly, in general, we observed no significant survival differences between our cohort of advanced multifocal SI-NETs and our control cohort of advanced unifocal SI-NETs. This finding is consistent with results from previous studies on SI-NETs [14,19,29,30]. In line with this general finding, intertumoral variation of WHO grade, SSTR2, serotonin, or CDX2 did not impact patient survival within the group of multifocal tumors or in comparison to unifocal NETs, pointing towards an, at best, limited prognostic relevance of morphological heterogeneity in multifocal SI-NETs and even of multifocality in general. Therefore, our data indicate that no specific prognostic information is missed when resection specimens of multifocal SI-NETs receive a histopathological diagnostic workup that is generally similar to unifocal SI-NETs. In particular, extensive analyses of the central immunohistochemical parameters of all synchronous lesions that go beyond the leading primary tumor appear to be unnecessary.

Our study was limited by the fact that our analyses were retrospective in nature and by the limited size of both cohorts with relatively few events in the cohort of multifocal SI-NETs. Therefore, additional studies in larger cohorts are needed to confirm our results from this study collective. Another limitation of our work is that we do not (yet) know the genomic background of each neoplasm of our cohort so that we have no molecular data that might deliver specific explanations for the diverse intertumoral heterogeneity that we morphologically observed in our cohort. Finally, our current study did not tackle the very interesting topic of heterogeneity of the multifocal primaries compared to metastatic lesions (e.g., regional lymph nodes, peritoneal deposits, and hepatic metastases), which should also be investigated in future studies.

## 5. Conclusions

In conclusion, our in-depth morphological analyses identified considerable and diverse intertumoral heterogeneity of central diagnostic immunohistochemical parameters within multifocal SI-NETs. However, our study highlights that these intertumoral differences are of no prognostic relevance and therefore do not justify a mandatory extended histopathological workup in daily clinicopathological practice. In particular, WHO grade, the most important histopathological parameter in neuroendocrine neoplasms, can be adequately assessed when the most advanced primary tumor is evaluated for its proliferative activity.

## Figures and Tables

**Figure 1 cancers-14-03963-f001:**
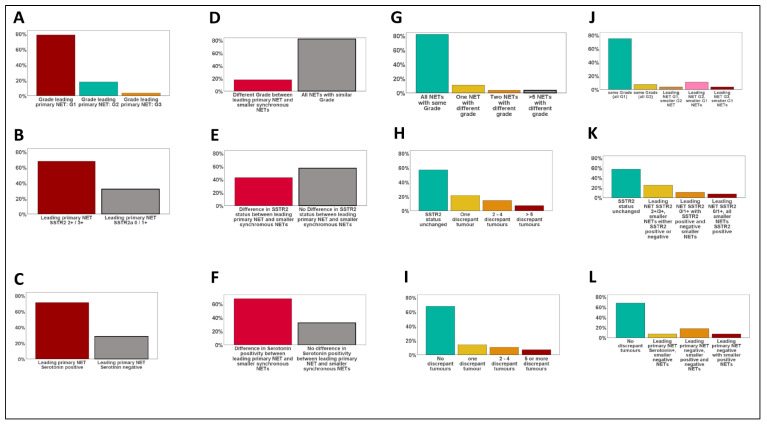
Heterogeneity of morphologic parameters in multifocal SI-NETs. (**A**–**C**): Frequency of WHO grades G1–G3 (**A**) and SSTR2 expression groups (**B**) as well as serotonin expression within the leading primary NET. (**D**–**F**): Frequency of changes of WHO grade (**D**), SSTR2 expression (**E**), and serotonin positivity between the leading primary NETs and synchronous smaller NETs. (**G**–**I**): Number of divergent NETs compared to the leading primary for WHO grade (**G**), SSTR2 status (**H**), and serotonin expression (**I**). (**J**–**L**): Type of change between the leading primary NET and the smaller synchronous tumors for WHO grade (**J**), SSTR2 (**K**), and serotonin (**L**).

**Figure 2 cancers-14-03963-f002:**
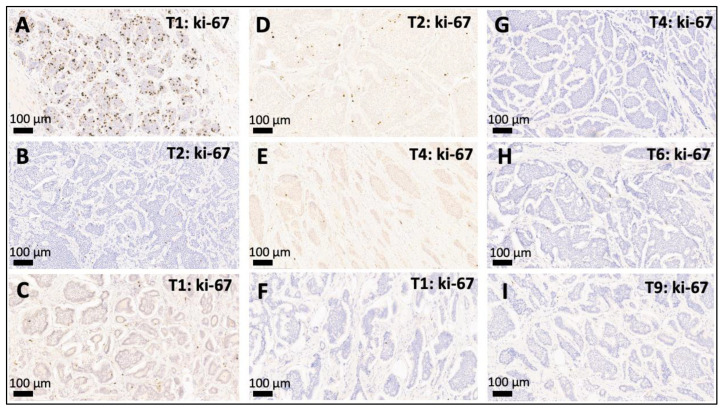
Intertumoral variation of WHO grade (G1–G3) between multifocal SI-NETs. (**A**,**B**): Discordant multifocal SI-NET with a G3 leading primary NET with a proliferation activity of 21% (**A**) accompanied by a smaller synchronous G1 NET with a proliferation rate of <1% (**B**). (**C**–**E**): Discordant multifocal SI-NET with a G1 leading primary NET ((**C**), proliferation < 1%) accompanied by one smaller synchronous G2 NET with a slightly elevated proliferation rate of 3.5% (**D**) and another G1 SI-NET (**E**). (**F**–**I**): Example of a concordant multifocal SI-NET where the leading primary G1 NET ((**F**), proliferation < 1%) is accompanied by multiple smaller lesions that are also G1 (**G**–**I**), proliferation < 1%). Depicted are the leading primary NET and selected smaller synchronous jejunoileal NETs (20×) from patients with multifocal SI-NETs: T1: leading primary NET, T*: number of the respective smaller synchronous NET within the same patient. WHO grade: G1 (ki-67 index < 3%), G2 (ki-67 index 3–20%), and G3 (>20%).

**Figure 3 cancers-14-03963-f003:**
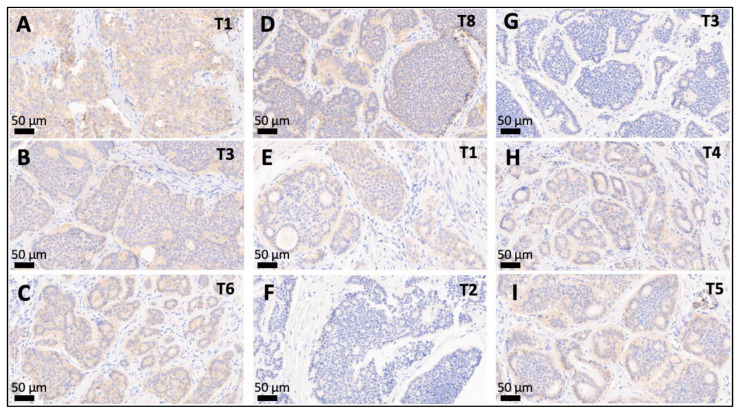
Intertumoral variation of SSTR2 status between multifocal SI-NETs. (**A**–**D**): Multifocal SI-NET with an SSTR2 3+ leading primary NET (**A**) accompanied by a smaller SSTR2 2+ synchronous NETs (**B**–**D**). (**E**–**H**): Discordant multifocal SI-NET with an SSTR2 2+ leading primary NET ((**C**), proliferation < 1%) accompanied by two smaller synchronous SSTR2-negative NETs (**F**,**G**), and another two SSTR2 2+ NETs (**H**,**I**). Depicted are the leading primary NET and selected smaller synchronous jejunoileal NETs (20×) from patients with multifocal SI-NET: T1: leading primary NET, T*: number of the respective smaller synchronous NET within the same patient. Abbreviations: SSTR2 = somatostatin receptor 2a.

**Figure 4 cancers-14-03963-f004:**
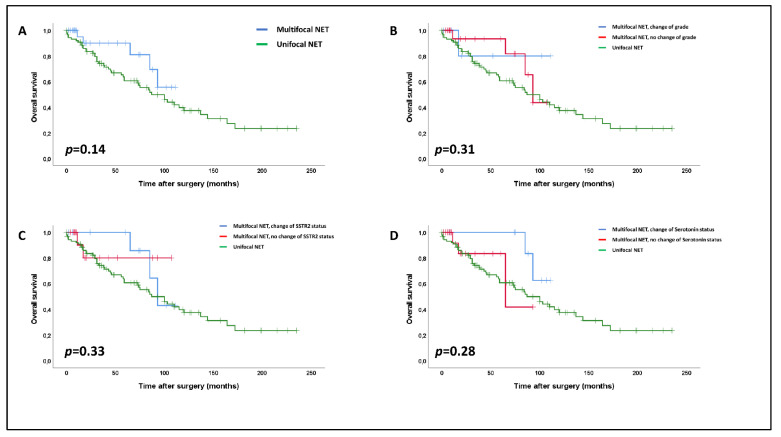
Impact of multifocality (**A**) and of intertumoral heterogeneity regarding the WHO-grade (**B**), SSTR2 status (**C**) and serotonin expression (**D**) on overall survival within multifocal SI-NETs compared to unifocal SI-NETs (log-rank rest). The vertical bars in the Kaplan–Meier curves show censored patients at the respective time points.

## Data Availability

The data is presented in this study are available in this article (and Supplementary Material).

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
