# Peer review of "Frequency and Prognostic Significance of Intertumoural Heterogeneity in Multifocal Jejunoileal Neuroendocrine Tumours"

_cancers, 2022, doi:10.3390/cancers14163963_

Round 1
Reviewer 1 Report
This is a very interesting study evaluating the concept of intertumoral heterogeneity in patient with multifocal SI-NETs. Authors did a great job providing a sound analysis plan and analyzing their findings. I do not have any specific comments to make. For the future, it would be interesting to evaluate the genomic landscape of multifocal SI-NETs in that setting.
Reviewer 2 Report
The article by Jesinghaus et al is exploring intertumoral heterogeneity in multifocal SI-NETs
and whether it affects the prognosis of the disease. The article is well written and easy to
read. It is an interesting finding that there is such a heterogeneity regarding important SINET
IHC markers in different PTs within patients. A question that is immediately raised is:
what is the situation in the metastases? Is this heterogeneity reflected in the metastases?
Since the primary tumours most often are surgically removed, the following treatment is
directed against the remaining metastases. Therefore, it would be of great interest to
explore if the situation is similar here.
The authors´ conclusion of the study is that an eventual heterogeneity between multiple
primary tumours does not affect the prognosis of the disease. Therefore, one should not
treat patients with multifocal PTs differently than ones with unifocal PTs. Most patients in
the study had a stage 4 disease. As the authors mention, tumour stage is a well-known
prognostic factor. However, there is very little discussion on the characteristics of the
metastases in the patients. Were any of the metastases examined pathologically? It would
improve the interpretation of whether the PT heterogeneity is of any importance if an
analysis of the metastases could be presented. Please comment.
Regarding survival analysis and Figure 3:
The authors perform a survival analysis between patients with unifocal and multifocal
primaries. However, very few patients seem to have died during follow-up, i.e. there were
very few “true” events (= deaths). In fact, the readers are not told all the figures: How many
in the “unifocal group” died? Only disease-specific deaths are presented in the “multifocal
group”. Were there patients that died from other reasons? Furthermore, the small cohorts
are another weakness for the interpretation of the data. The authors should comment on
this as limitations of the study.
Censored patients are left-out Figure 3. The authors should consider adding censored
patients in the graph. Also, a table with patients at risk would improve the understanding.
Minor comment:
Line 304: the authors present an explanation model for why primary tumours often are not
visible on SSTR imaging. However, the most reasonable explanation is that due to their small
size (compared to mets) the radionuclide uptake is not high enough to detect them.
Minor adjustments:
Line 120: please specify which serotonin antibody was used.
Line 228: Sentence beginning with “In a first general analysis ..” is hard to read and
understand. What were the observations? Please rewrite.
Line 297: It seems like a “to” is missing in the sentence. ..the pathologist who has to consider
that..
